# Prevalence and risk factors associated with rural women's protected against tetanus in East Africa: Evidence from demographic and health surveys of ten East African countries

**Alebachew Taye Belay** *, **Setegn Much Fenta**, **Setegn Bayabil Agegn, Mitiku Wale Muluneh**

Department of Statistics, Debre Tabor University, Debre Tabor, Ethiopia

* alex016stat@gmail.com

## Abstract

**Data Availability Statement:** The survey datasets used in this study were based on a publicly available dataset that is freely available online with

### Background

Tetanus is a deadly bacterial infection caused by Clostridium tetani wound contamination characterized muscular spasms and autonomic nervous system dysfunction. Maternal and neonatal tetanus occurs under improper hygiene practices during childbirth. Globally, an estimated 3.3 million newborn deaths occur every year, and about 9,000 babies die every day in the first 28 days of life. This study sought to identify risk factors associated with the immunization of rural women against tetanus in rural areas in ten East African countries.

### Method

The data used in this study were taken from the Demographic and Health Survey (DHS) of ten East African countries (Ethiopia, Burundi, Comoros, Zimbabwe, Kenya, Malawi, Ruanda, Tanzania, Uganda and, Zambia). Multivariable binary logistic regression is used to determine the risk factors associated with tetanus-protected women in east Africa.

### Results

The weighted total samples of 73735 rural women were included in the analysis. The combined prevalence of tetanus immunization among protected rural women in ten East African countries was 50.4%. Those women with age of 24–34 (AOR = 0.778; 95%CI: 0.702–0.861), higher educational level (AOR = 4.010; 95%CI: 2.10–5.670), rich women (AOR = 3.097;95%CI: 2.680–3.583), mass media coverage (AOR = 1.143; 95%CI: 1.030–1.269), having above three antenatal care follow up (AOR = 1.550; 95% CI: 1.424–1.687), big problem of distance to health facility (AOR = 0.676; CI: 0.482–0.978) and place of delivery health facility (AOR = 1.103; 95% CI: 1.005–1.210) had a significant effect on women's protected from tetanus.

no participant's identity from http://www.
dhsprogram.com/data/available-datasets.cfm.
Approval was sought from MEASURE DHS/ICF
International and permission was granted for this
use.

**Funding:** Funding was not provided for this study.

**Competing interests:** The authors state that there
is no conflict of interest.

Abbreviations: AOR, Adjusted Odds Ratio; AIC,
Akaka's Information Criterion; BIC, Bayesian's
Information Criterion; CI, Confidence Interval; DHS,
Demographic Health Survey; EA, East Africa; MT,
Maternal Tetanus; NT, Neonatal Tetanus; SPSS,
Statistical Package for Social Science; TT, Tetanus;
WHO, World Health Organization.

## Conclusion

The coverage of tetanus immunization in East Africa was very low. Public health programs
target rural mothers who are uneducated, poor households, longer distances from health
facilities, mothers who have the problem of media exposure, and mothers who have not
used maternal health care services to promote TT immunization.

## Background

Tetanus is a deadly bacterial infection caused by Clostridium tetani wound contamination and
is characterized by muscular spasms and autonomic nervous system dysfunction [1, 2]. Mater-
nal and neonatal tetanus occurs under improper hygiene practices during childbirth [1]. Teta-
nus, which occurs during pregnancy or within 6 weeks of the end of pregnancy, is called
maternal tetanus (MT) and neonatal tetanus (NT) occurs in the first 28 days of life [1].

Globally, an estimated 3.3 million newborn deaths occur every year, and about 9,000 babies
die every day in the first 28 days of life. Of this death, in 2015, the World Health Organization
(WHO) estimates that 34,019 infants died due to NT, [2] and also, in 2017, approximately
30,848 neonates died of neonatal tetanus [3, 4]. It is estimated that tetanus deaths in 2018 were
25, 000 [5]

Despite the World Health Organization (WHO) initiatives to eliminate tetanus, it continues
to lead to significant maternal and neonatal deaths [6]. In the poorest parts of the world, neo-
natal tetanus (NT) is responsible for 14% of neonatal deaths, while maternal tetanus (MT) is
responsible for at least 5% of maternal deaths [7].

In general, tetanus mortality tends to be high in the absence of medical treatment where
case fatality approaches 100%; this percentage decreases to 10%–60% in the presence of hospi-
tal care, depending on the availability of intensive care facilities [8, 9].

Similarly, the majority of the Sub-Saharan African countries could hardly reach the TT
immunization target set to be covered [10], in sub-Saharan Africa, which accounts for nearly
half of the global maternal and neonatal mortality by tetanus (6).

Evidence shows that a spread of determinant factors affects the use of TT protection. For
instance, women's education and wealth index might impose variations in immunization cov-
erage [11]. Furthermore, studies indicate that maternal age, marital status and mother's occu-
pation, distance from health facilities, range of visits to healthcare facilities, and also the
number of children within the house can even significantly verify TT immunization usage.
Furthermore, the findings of different studies have shown that antenatal care visits (ANC vis-
its) [10], women's education [12, 13], income [13, 14], distance to a health facility [13, 14], resi-
dence[15], maternal age at first birth [16], women's employment status [17] and media
exposure [18] are significantly associated with mothers who protected TT immunization
against tetanus.

Maternal vaccination is an important strategy to prevent maternal, neonatal, and infant dis-
eases [19–21]. Despite the evidence on the safety and effectiveness of vaccines for mothers,
there are still challenges in achieving high vaccination coverage during pregnancy worldwide
[22].

Some studies have shown that 94% of the reduction in neonatal mortality can be obtained
by immunization against tetanus (TT) in pregnant mothers [29] and mothers of reproductive
age with at least two doses of TT. As a result, tetanus vaccines are recommended to eliminate
maternal and neonatal tetanus in these areas of the community [23, 24].

Moreover, the previous studies did not use a multi-country method to identify factors associated with tetanus immunization in rural women based on the pooled Demographic and Health Survey (DHS) data in east Africa.

The problem of TT is more pronounced in developing countries like African states, as they have a significant number of the poorest and most neglected population groups that have little or no access to medical care [25]. Despite the problem being common in Africa, to the best of our knowledge, there is a scarcity of studies that determine the factors associated with a mother's protected from TT immunization against maternal and neonatal tetanus. Therefore, this study was designed to show the prevalence and investigate the risk factors associated with tetanus protected from rural mothers in rural areas of ten East African countries. Carrying out this research is important for implementing prevention strategies for maternal tetanus by highlighting the risk factors associated with TT vaccination. Finally, this study aimed to identify the determinants of tetanus vaccination among rural women in east Africa.

## Methods

### Study area and data source

The data used in this study were obtained from the Demographic and Health Survey (DHS) of ten East African countries (Ethiopia, Burundi, Zimbabwe, Kenya, Malawi, Ruanda, Tanzania, Comoros, Uganda, and Zambia). Countries were selected based on geographical location, adjacency, and availability of data on the outcome variable. The DHSs were a nationally representative survey that collects data on basic health indicators like mortality, morbidity, family planning service utilization, fertility, maternal and child vaccination. The data were derived from the measure DHS program (https://www.dhsprogram.com/Data). It also collects standard protocols from most low and middle-income countries to facilitate comparability between countries. Sample selection in the surveys was based on a two-step stratified sampling method. Each country was divided into clusters. In the first phase, Enumeration Areas (EI) were selected in each group and a household listing exercise was conducted in all selected enumeration areas. The household list was used as the basis for household selection. During the second step, households were chosen from each enumeration area. In this study, we used the "latest" or most recent surveys conducted from 2014 to 2019, and the data used for analysis were obtained by pooling the DHS data of the ten East Africa countries. The data were extracted from each country and merge in to one sample data for the analysis. The combined DHS data included 73735 rural women who participated in the tetanus vaccination questionnaire. Therefore, the analysis of the study was based on 73735 samples (Table 1).

### Variables of the Study

The outcome variable for this study was rural mothers' who were protected against tetanus in ten EA countries, which was dichotomized as mothers protected from TT and not protected from TT. In this study, a woman is said to be protected from tetanus before birth when women receive at least two TT injections during pregnancy for her most recent birth, or two or more injections (the last within 3 years of the most recent birth), or three or more injections (the last within 5 years of the most recent birth), or four or more injections (the last within 10 years of the most recent birth), or five or more injections at any time before the most recent birth [26].

The independent variables for this study were the educational status of women, age of the mother at first birth, costs covered by health insurance, occupation, marital status, husband/partner's age, wealth index, mother's pregnancy wanted, mothers had a terminated pregnancy, antenatal care (ANC), Place of delivery, distance from health facility and media exposure.

**Table 1. Selected number of study participants in the 10 East Africa selected countries' using the demographic and health surveys 2014–2019.**

| List of east Africa selected Country | Participants selected from the target population |
| --- | --- |
| Ethiopia | 7043 |
| Burundi | 8655 |
| Comoros | 6371 |
| Zimbabwe | 4749 |
| Kenya | 7119 |
| Malawi | 13376 |
| Ruanda | 3904 |
| Tanzania | 7019 |
| Zambia | 7296 |
| Uganda | 8203 |

Media exposure is created from four variables from DHS data called newspaper reading, radio and television viewing, and telephone usage. Then it is recorded as yes if a woman had used at least one of the four media sources and if a woman didn't have any of the three media sources is said to be no.

## Data analysis procedure

The data were extracted using SPSS version 25 statistical software to derive important variables and relevant inferences from the DHS data. The data were analyzed using R version 4.0.3. Descriptive statistics, such as percentages, bar graphs, and frequency tables, were used to describe the respondent study variables. This study also used a combination of the chi-square test to determine whether the response variable was associated with different cofactors. Moreover, a multivariable binary logistic regression model must correspond to a response variable with two categories (mothers protected against TT by TT immunization with two categories yes or no). A binary logistic regression model was used to determine the risk factors for TT-protected mothers using tetanus vaccination TT2+.

The risk factor outcome was reported in terms of an adjusted rating ratio with a significance level of 5% (95% CI). In the univariate analysis, a significance level of 25% was considered a candidate for the multivariate analysis of data analysis. All variables with p values $\leq 0.05$ were considered statistically significant.

For a binary response $Y_i$ and a quantitative explanatory variable $X_{ij}$, j = 1, 2 . . . M and I = 1, 2 . . . N, let $\pi_i = P(X_{ij})$ denote the "success probability" when $X_{ij}$ takes the values $X_{ij}$. The problem with the linear model is that the probability model E(Y) is used to approximate a probability value $\pi i = P(Y_i = 1)$ within the interval 0 and 1, while $E(Y_i)$ Is not constrained. Therefore, we apply the logit transformation where the transformed quantity lies in the interval from minus infinity to positive infinity and it is modeled as

$$\text{The logit } (\pi_i) = \log \left( \frac{\pi i}{1 - \pi i} \right) = \alpha + \beta_1 X_1 + \beta_2 X_2 + \ldots, + \beta_P X_P$$

$\beta_i$ = the coefficient of the i[th] predictor variable determines the rate of increase or decrease of $X_{ij}$ On the log of the odds that $Y_i = 1$, controlling for the other X's [27].

The linkage function used for binary logistic regression modeling was legit, probit complementary log-log, and negative log-log function that fits the data, which are described as follows (Table 2):

**Table 2. Link function for the logistic regression model.**

| Function | Form | Typical application |
|---|---|---|
| Legit | Log $\left[\frac{\pi(X_i)}{1-\pi(X_i)}\right]$ | Evenly distributed categories |
| Complementary log-log | Log [-Log (1- $\pi$ ($X_i$))] | Higher categories are more probable |
| Negative log-log | -Log $\pi$ ($X_i$) | Lower categories are more probable |
| Probit | $\phi^{-1}$ [$\pi$ ($X_i$)] | Normally distributed latent variable |

## Model selection

Akaike's information criterion (AIC) and Bayesian information criterion (BIC) were used to compare the candidate link function of models. The model with the minimum value of the information criterion is chosen as the best link for the analysis [34].

## Ethical consideration

The study was used secondary data analysis of publicly available survey data from the DHS program, ethical approval, and participant consent was not necessary for this study. We requested the DHS program, and permission was granted to download and use the data for this study from http://www.dhsprogram.com. There is no name of individuals or households addresses in the data files. Therefor ethical approval was not necessary for this study.

## Results

The combined prevalence of rural women protected against tetanus before birth in the ten East African countries was (50.4%). While more than half of the rural women 49.6% had no tetanus protection (Fig 1).

   The prevalence of protected rural women from tetanus in ten EA countries was Ethiopia (63.1% not protected and 36.9% protected), Burundi (53.8% not protected and 46.2% protected), Comoros (50.3% not protected and 49.7% protected), Kenya (55.0% not protected and 45.0% protected), Malawi (51.3% not protected and 48.7% protected), Rwanda (35.0% not protected and 65.0% protected), Tanzania (48.3% not protected and 51.7% protected), Zambia (48.6% not protected and 51.4% protected), Uganda (40.2% not protected and 59.8% protected) and Zimbabwe (39.8% not protected and 60.2% protected) (Fig 1).

   More than half of women were not protected from tetanus in Ethiopia (63.1%), Kenya (55.0%), Burundi (53.8%), and Malawi (51.3%), and Comoros (50.3%) whereas Uganda (40.20%), Zimbabwe (39.80%), and Rwanda (35%) have below 50% prevalence of women's not protected from tetanus (Fig 1).

### Socio-demographic characteristics of respondents

The majority of 33676 (45.7%) of respondents' mothers were aged between 25 and 34 years and 38857 (53.0%) were primary education. More than three-fourths 55603 (75.4%) of the sex of household heads were male, and 33135 (44.9%), 26206 (35.5%) and 14394 (19.5%) respondents were poor, middle, and rich respectively (Table 2).

   More than three-fourths of respondents' immunizations were not covered by health insurance, 64914 (88.0%) and the majority 45760 (65.5%) of respondents were married. More than half of respondents 8705 (57.5%) were told about pregnancy complications and 43270 (58.7%) had no media coverage (Table 2).

   More than half of respondents 39996 (54.2%) were more than three ANC visits, 4675 (6.3%) were no ANC visit, and 29064 (39.4) were one have to three ANC visits, and also three

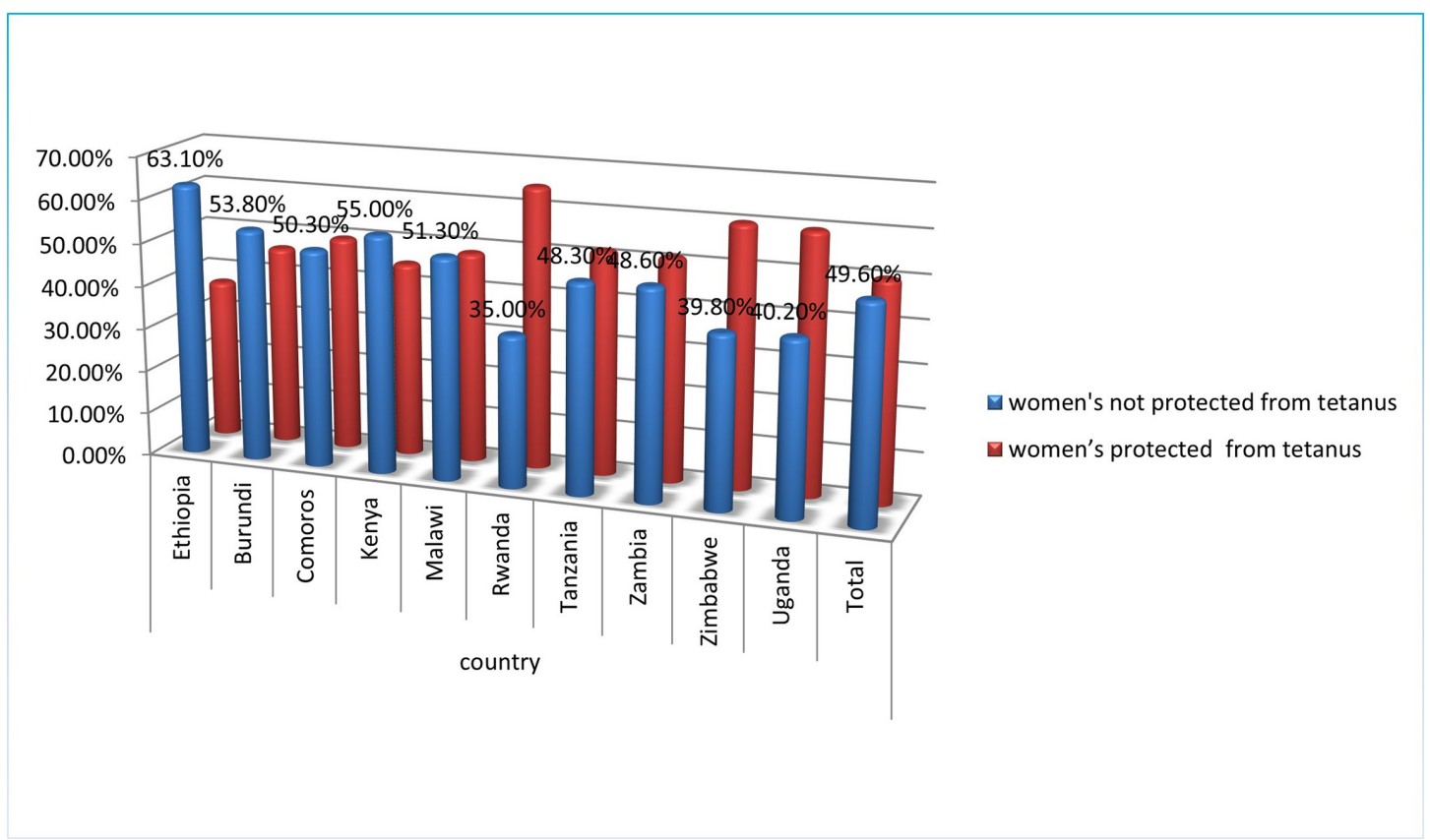

**Fig 1. Percentage of protected women from tetanus before birth in East Africa (EA) countries.**

fourth of respondents 56382 (76.5) were getting immunization to the health facility. Finally, more than three fourth women's 6482 (82.3%) had a big problem of distance to health facilities whereas the remaining 8821 (17.7%) women's haven't a big problem of distance to health facilities.

Moreover, the chi-square test of association showed that respondent's current age, educational level, sex of household head, wealth index, distance to health facilities, ever had a terminated pregnancy, covered by health insurance, current marital status, respondent's occupation, told about pregnancy complications, mass media coverage, ANC visit, place of delivery, and country were significantly correlated TT protection amongst east Africa rural women but currently pregnancy and mother pregnancy wanted were no associated with women protected from tetanus (Table 3).

### Selection of link function

Since the candidate link function of the data complementary log-log had a small value of AIC (13410.37) and BIC (13635.67). Therefore, the binary logistic regression model was fitted using a complementary log-log link function (Table 4).

### Multivariable analysis

In the multivariable logistic regression analysis; respondent's current age, educational level, wealth index, marital status, women's occupation, told about pregnancy complications, mass

**Table 3. Socio-demographic, economic, and maternal characteristics of respondents in ten East African countries from 2014–2019.**

| Variables | Tetanus protected women before birth | | | $x^2$ value (P-value) |
|---|---|---|---|---|
| | No (%) | Yes (%) | Total (%) | |
| Age of mother | | | | 511.342 (<0.0001) |
| Below 24 years | 9978 (44.2) | 12602 (55.7) | 22580 (30.6) | |
| Age between 25 to 34 | 16888 (50.1) | 16788 (49.9) | 33676 (45.7) | |
| Age above 35 | 9700 (55.5) | 7779 (44.5) | 17479 (23.7) | |
| The highest educational level of mother | | | | 3374.514 (<0.0001) |
| No education | 7078 (56.7) | 5398 (43.3) | 12476 (17.0) | |
| Primary | 18529 (47.7) | 20328 (52.3) | 38857 (53.0) | |
| Secondary | 9437 (49.3) | 9688 (50.7) | 19125 (26.1) | |
| Higher | 1172 (42.0) | 1616 (58.0) | 2788 (3.8) | |
| Sex of household head | | | | |
| Male | 27876 (50.1) | 27727 (49.9) | 55603 (75.4) | 26.658 (< 0.0001) |
| Female | 6982 (47.9) | 9442 (52.1) | 18132 (24.6) | |
| Wealth index | | | | |
| Poor | 17582 (53.1) | 15553 (46.9) | 33135 (44.9) | 317.311 (<0.0001) |
| Middle | 12002 (45.8) | 14204 (54.2) | 26206 (35.5) | |
| Rich | 6240 (48.5) | 7412 (51.5) | 14394 (19.5) | |
| Age of respondent at 1st birth | | | | 15.436 (<0.0001) |
| Age below 24 years | 33767 (49.8) | 34075 (50.2) | 67842 (92.0) | |
| Age between 25–30 years | 2486 (48.0) | 2694 (52.0) | 5180 (7.0) | |
| Age above 30 years | 313 (43.9) | 400 (56.1) | 713 (1.0) | |
| Currently pregnant | | | | 0.231 (0.217) |
| No or unsure | 33005 (49.6) | 33537 (50.4) | 66542 (90.2) | |
| Yes | 3561 (49.5) | 3632 (50.5) | 7193 (9.8) | |
| Mother pregnancy wanted | | | | 3.398 (0.138) |
| Then | 2058 (49.8) | 2078 (50.2) | 4136 (6.2) | |
| Later | 31580 (50.7) | 30768 (49.3) | 62348 (93.6) | |
| Ever had a terminated pregnancy | | | | 10.73 (0.0001) |
| No | 31441 (49.4) | 32145 (50.6) | 63586 (86.2) | |
| Yes | 5124 (50.5) | 5024 (49.5) | 10148 (13.8) | |
| Covered by health insurance | | | | 8.234 (<0.004) |
| No | 32318 (49.8) | 32596 (50.2) | 64914 (88.0) | |
| Yes | 4248 (48.2) | 4573 (51.8) | 8821 (12.0) | |
| Current marital status | | | | 385.345 (<0.0001) |
| Single | 2081 (45.0) | 2544 (55.0) | 4625 (6.6) | |
| Married | 24271 (53.0) | 21489 (47.0) | 45760 (65.5) | |
| Living with partner | 4601 (47.7) | 5053 (52.3) | 9654 (13.8) | |
| Widowed | 1880 (42.1) | 2582 (57.9) | 4462 (6.4) | |
| Divorced | 2421 (45.4) | 2909 (54.6) | 5330 (7.6) | |
| Respondent's occupation | | | | 510.571 (<0.0001) |
| Housewife | 11031 (52.2) | 10107 (47.8) | 21138 (28.7) | |
| Government employed | 2202 (36.1) | 3904 (63.9) | 6106 (8.3) | |
| Other type of worker | 23333 (50.2) | 23158 (49.8) | 46491 (63.1) | |
| Told about pregnancy complications | | | | 200.57 (<0.0001) |
| No | 3154 (49.0) | 3286 (51.0) | 6440 (42.5) | |
| Yes | 3262 (37.5) | 5443 (62.5) | 8705 (57.5) | |

(*Continued*)

**Table 3.** (Continued)

| Variables | Tetanus protected women before birth | | | $x^2$ value (P-value) |
| --- | --- | --- | --- | --- |
| | No (%) | Yes (%) | Total (%) | |
| Mass media coverage | | | | 1232.466 (<0.0001) |
| No | 23805 (55.0) | 19465 (45.0) | 43270 (58.7) | |
| Yes | 12761 (41.9) | 17704 (58.1) | 30465 (41.3) | |
| ANC visit | | | | 3791.908 (<0.0001) |
| No ANC visit | 4185 (89.5) | 490 (10.5) | 4675 (6.3) | |
| One have to three ANC visit | 15228 (52.4) | 13836 (47.6) | 29064 (39.4) | |
| More than three ANC visit | 17153 (42.9) | 22843 (57.1) | 39996 (54.2) | |
| Place of delivery | | | | 1445.17 (<0.0001) |
| Health facility | 25771 (45.7) | 30611 (54.3) | 56382 (76.5) | |
| Home and other traditional place | 10795 (62.2) | 6558 (37.8) | 17353 (23.5) | |
| Distance to health facilities | | | | 1235.466 (<0.0001) |
| Big problem | 32318 (49.8) | 32596 (50.2) | 6482 (82.3) | |
| No big problem | 4248 (48.2) | 4573 (51.8) | 8821 (17.7) | |

media coverage, place of delivery, age of respondent at 1st birth, country, distance to health facility and number of ANC visits remained statistically significantly associated with women's protected from tetanus (Table 5).

Hence, mothers' current age between 24 to 34 were 0.830 times less likely to have been women's protected from tetanus than mothers' age below 24 (AOR = 0.830; 95%CI: 0.778, 0.886). In addition, mother's current age above 34 was 0.733 times less likely to have been women's protected from tetanus than mother's age under 24 (AOR = 0.733; 95%CI: 0.673, 0.799); which shows that at the age of mother increase the rural women's protected from tetanus become decrease.

Additionally, the odds of having a woman protected from tetanus were 0.792 less likely than women's who had primary education compared to no education (AOR = 0.792; 95%CI: 0.737, 0.852); and a woman protected from tetanus were 0.833 times less likely among mothers who have secondary educational levels compared to no education (AOR = 0.833; 95%CI: 0.751, 0.922) Similarly, mothers with higher education were 1.276 times more likely to be women protected against tetanus than those without education (AR = 1.276; 95% CI: 1.092, 1.490).

Rural mothers from middle wealth households were a 1.976 (AOR = 1.976; 95% CI: 1.802, 2.166) times higher probability of women being protected from tetanus compared to poor households. Similarly, rich wealthiest households were a 1.589 (AOR = 1.589; 95% CI: 1.425, 1.771) times higher probability of women being protected from tetanus compared to a poor household.

Being current marital status of a mother who lives with a partner was 0.780 times less likely to have women's protected from tetanus than the mother's marital status was single

**Table 4. Candidate link function for binary logistic regression.**

| Information criteria | Type of link function | | | |
| --- | --- | --- | --- | --- |
| | Logit | Probit | Complimentary log | Negative log-log |
| AIC | 13454.97 | 13446.93 | 13410.37 | 13456.48 |
| BIC | 13680.28 | 13672.23 | 13635.67 | 13798.42 |

**Table 5. Risk factors associated with rural women's protection from tetanus in 9 East Africa countries from 2014–2019.**

| Variables | COR (95% CI) | P_value | AOR (95% CI) | P_value |
|---|---|---|---|---|
| Constant | | | 0.395 (0.319, 0.488) | < 0.0001 |
| Age of mother (ref = Below 24 years) | | | | |
| Age between 24 to 34 | 0.787 (0.761, 0.814) | < 0.0001 | 0.830 (0.778, 0.886) | < 0.0001 |
| Age above 35 | 0.635 (0.610, 0.661) | < 0.0001 | 0.733 (0.673, 0.799) | < 0.0001 |
| Educational level (ref = no education) | | | | |
| Primary | 1.439 (1.381, 1.498) | < 0.0001 | 0.792 (0.737, 0.852) | < 0.0001 |
| Secondary | 1.346 (1.286, 1.409) | < 0.0001 | 0.833 (0.751, 0.922) | 0.0005 |
| Higher | 1.823 (1.672, 1.991) | < 0.0001 | 1.276 (1.092, 1.490) | 0.0014 |
| Sex of household head (ref = male) | | | | |
| Female | 1.030 (1.062, 1.134) | < 0.0001 | 1.091 (0.966, 1.097) | 0.3615 |
| Wealth index (ref = poor) | | | | |
| Middle | 1.414 (1.371, 1.467) | < 0.0001 | 1.976 (1.802, 2.166) | < 0.0001 |
| Rich | 1.191 (1.140, 1.241) | < 0.0001 | 1.589 (1.425, 1.771)* | < 0.0001 |
| Ever had a terminated pregnancy (ref = no) | | | | |
| Yes | 0.930 (0.891, 0.974) | 0.0028 | 0.965 (0.882, 1.055) | 0.4395 |
| Covered by health insurance (ref = no) | | | | |
| Yes | 1.124 (1.074, 1.171) | 0.00412 | 0.924 (0.844, 1.012) | 0.0881 |
| Current marital status (ref = Single) | | | | |
| Married | 0.721(0.676, 0.774) | < 0.0001 | 1.038 (0.911, 1.185) | 0.5786 |
| Living with partner | 0.801(0.745, 0.873) | 0.0028 | 0.780 (0.643, 0.944) | 0.0111 |
| Widowed | 0.698(0.641, 0.795) | 0.0060 | 0.941 (0.740, 1.188) | 0.6137 |
| Divorced | 0.976(0.102, 1.063) | 0.6690 | 1.055 (0.902, 1.236) | 0.4995 |
| Mother occupation (ref = Housewife) | | | | |
| Government employed | 2.014 (1.881, 2.165) | < 0.0001 | 0.865 (0.762, 0.980) | 0.0231 |
| other type of worker | 1.053 (1.021, 1.084) | < 0.0001 | 0.968 (0.914, 1.025) | 0.2650 |
| Told about pregnancy complications (ref = no) | | | | |
| Yes | 1.605 (1.501, 1.721) | < 0.0001 | 1.103 (1.039, 1.170) | 0.0010 |
| Mass media coverage (ref = no) | | | | |
| Yes | 1.751 (1.691, 1.802) | < 0.0001 | 1.968 (1.530, 2.569) | < 0.0001 |
| ANC visit (ref = no ANC visits) | | | | |
| One have to three visit | 8.032 (7.271, 8.891) | < 0.0001 | 1.113 (1.125, 1.313) | 0.0041 |
| more than 3 ANC visit | 11.38(10.312,12.59) | < 0.0001 | 1.324 (1.252, 1.400) | 0.0010 |
| place of delivery (ref = home and other traditional places) | | | | |
| health facility | 2.072 (1.991, 2.144) | < 0.0001 | 1.165 (1.012, 1.332) | 0.0458 |
| Distance to a health facility (ref = big problem) | | | | |
| No big problem | 0.653 (1.018, 1.081) | < 0.0001 | 0.678 (0.483, 0.979) | < 0.0001 |
| Age of household head | 0.995 (0.991, 0.105) | | 0.100 (0.996, 1.004) | 0.5689 |
| Age of respondent at 1st birth (ref = Below 24) | | | | |
| Between 24–30 | 1.125 (1.056, 1.182) | 0.0135 | 1.206 (1.092, 1.330) | < 0.0001 |
| Above 30 | 1.326 (1.147, 1.542) | 0.0019 | 1.416 (1.097, 1.808)* | < 0.0001 |
| Country (ref = Ethiopia) | | | | |
| Burundi | 1.467 (1.376, 1.564) | < 0.0001 | 1.323 (1.134, 1.545) | 0.0004 |
| Comoros | 1.685 (1.57,3 1.805) | < 0.0001 | 1.324 (1.161, 1.512) | < 0.0001 |
| Kenya | 1.398 (1.307, 1.495) | < 0.0001 | 0.908 (0.788, 1.047) | 0.1813 |
| Malawi | 1.619 (1.526, 1.718) | < 0.0001 | 2.370 (2.083, 2.702) | < 0.0001 |
| Rwanda | 3.171 (2.923, 3.442) | < 0.0001 | 1.542 (1.337, 1.780) | < 0.0001 |
| Tanzania | 3.171 (2.923, 3.442) | < 0.0001 | 1.542 (1.337, 1.780) | < 0.0001 |

*(Continued)*

**Table 5.** (Continued)

| Variables | COR (95% CI) | P_value | AOR (95% CI) | P_value |
|---|---|---|---|---|
| Zambia | 1.828 (1.709, 1.956) | < 0.0001 | 2.217 (1.918, 2.567) | < 0.0001 |
| Zimbabwe | 1.806 (1.690, 1.931) | < 0.0001 | 1.232 (1.056, 1.438) | 0.0078 |
| Uganda | 2.584 (2.396, 2.787) | < 0.0001 | 0.711 (0.597, 0.846) | < 0.0001 |

(AOR = 0.780; 95% CI: 0.643, 0.944). Similarly, women who work government employed was 0.865 times less likely to have women's protected from tetanus than the women who were housewife (AOR = 0.865; 95% CI: 0.762, 0.980).

The odds of having women's protected from tetanus were 1.103 times higher among mothers who have to get information about pregnancy complications compared to no told about pregnancy complications (AOR = 1.103; 95%CI: 1.039, 1.170). Besides, mothers who had mass media coverage were 1.968 times more likely to have women's protected from tetanus compared to no mass media coverage (AOR = 1.968; 95% CI: 1.530, 2.569).

A mother who had one to three ANC visits was 1.113 (AOR = 1.113; 95% CI: 1.125, 1.313) times more likely to have women's protected from tetanus compared to a mother who did not have a PNC visit. Additionally, a mother who had more than three PNC visits was 1.324 (AOR = 1.324; 95% CI: 1.252, 1.400) times more likely to have women's protected from tetanus compared to a mother who did not have a PNC visit.

The tetanus immunization, health delivery was 1.165 (AOR = 1.165; 95% CI: 1.012, 1.332) times more likely to have women's protected from tetanus compared to home and other traditional places. Similarly, the odds of having women's protected from tetanus were 0.678 times lower among mothers who haven't a big problem of distance to the health facility compared to mother's in the household haven big problem to health facility (AOR = 0.678; CI: 0.483, 0.979).

Mother's age at 1st birth between 24 to 30 years was 1.206 times more likely to have women protected from tetanus than mother's age of 1st birth below 24 (AOR = 1.206; CI: 1.092, 1.330). Similarly, mothers' ages of 1st birth above 34 were 1.416 times more likely to have women's protected from tetanus compared to mothers' ages of 1st birth below 24 (AOR = 1.416; CI: 1.097, 1.808) (Table 5).

## Discussion

This study assessed the risk factor of women protected from tetanus among rural east Africa women's before birth. The study illustrated that the proportion of ten east Africa women's protected from tetanus toxoid protective immunization was found to be 50.4%. However, the proportion of this magnitude is low as compared to the study done [28] showed that TT2 + immunization coverage among pregnant mothers' were 75% worldwide, ranging from 95% in South East Asia to 53% in the East Mediterranean and 63% in Africa.

In this study, age of mother, educational level, wealth index, marital status, told about pregnancy complications, mass media coverage, ANC visit, place of delivery, and distance to the health facility, Age of mother at 1st birth, and Country was associated with mother protected from tetanus.

The odds of having women protected from tetanus among women aged above 24 years were higher than those women aged below 24 years. This result is consistent with a study conducted by different scholars [29–32]. This is because of the lack of information and education about the burden of tetanus and the importance of TT vaccination in the older age group compared with younger age groups of women [33]. This finding indicates the need of having

prioritization of adolescent vaccination as a necessary element of preventive health care to improve the health of their births and their health [34].

The other variable, the age of the mother at first birth had a significant effect on mothers protected from tetanus. This result contradicts the study was done [35], which showed that the age of the mother at first birth hadn't a significant effect on women's protection from tetanus. This might be the target population of this study were women who live in the rural area of East Africa, which have a problem of early marriage in this area. Therefore, the age of the mother at first birth may relate to early marriage that causes a knowledge gap on ANC visits and TT protective immunization.

The educational level of the mother was positively significantly associated with women's protection from tetanus; which is in line with other studies [30, 36], in this study, the increasing level of education significantly increases the mother's obtained TT immunization. Because education may strengthen the level of knowledge about the impact of TT on women and neonatal that increases the number of women who obtained TT immunization.

Another factor that is significantly associated with women's protection from tetanus in this study was the distance from a health facility; which is also consistent with other studies[10], in this study, mothers who perceive TT immunization may be due to no big problem of distance from a health facility; which increased the odds of having protected mother from TT immunization compared to women whose distance to a health facility as a big problem. This would be due to the reduced time, injury, and transportation costs associated with distance from the health facility. However, the other study showed that distance from health facilities has no significant effect on women's protection from tetanus [32, 35].

The access to mass media coverage increases the odds of women being protected from tetanus; which is in line with the other studies elsewhere [13, 18, 35]. In this study, women who have access to media exposures may strengthen knowledge about TT does immunization of women, which might be increased the odds of women being protected from tetanus. The other reason may be fewer media coverage in the rural area of most east Africa countries that affect knowledge of TT does immunization. However, the study contradicts other studies [32].

Another important factor that significantly affects women's protection from tetanus was the wealth index of women in the household, this study is also consistent with other studies conducted elsewhere[19]. In this study, the increasing wealth index of households from middle to rich causes to increase in the odds of women being protected from tetanus compared to the poor wealth index of households in the rural area of East Africa countries. This might be economic status has a significant impact on the use of health care services of the mother; rural areas most east Africa countries are far from health facilities that lead to households with low economic status could not afford the high transportation and maternity costs [14, 37]. Moreover, mothers who have low economic status might be busy with other activities to fulfill their human needs that lead mothers may not have enough time to utilize health care services compared with the high economic level in developing countries. However, this study contradicts the study done by [38] showed that the wealth index hadn't significant effect on women's protection from tetanus.

This study showed that ANC follow-up had a higher chance of having women's protected from tetanus compared to mothers with no ANC follow-up; which is also consistent with the previous study done in different countries worldwide [22, 25]. This might be that women with ANC follow up usually have increased awareness about the importance of taking TT immunization, one of the ANC service packages, and mothers who had ANC follow up are more likely to get vaccinated and immunized against tetanus which in turn results in births protected against neonatal tetanus [31].

Mothers who had information about pregnancy complications had increased the likelihood that women would be protected against tetanus, which is consistent with other studies

elsewhere. [13, 18]. In this study, women who have information about pregnancy complications may increase women's protection from tetanus, which might be rural mothers who have information about pregnancy complications may include the consequence tetanus for neonatal and mothers themselves if there is no ANC visit and TT immunization.

Marital status had significantly increased the odds of women being protected from tetanus, which in line with the previous study [35] showed that marital status had a significant effect on TT immunization, but the other study contradict with the study done [32] showed that marital status had no significant effect with women's protected from tetanus.

The other important variable countries had significantly affected the odds of women being protected from tetanus; this might be different geographical location may increase the knowledge gap about immunization. Similarly, had significantly affected the odds of women being protected from tetanus; it may be different implications for immunization-based religious beliefs.

The variable place of delivery had significantly increased the odds of women's protection from tetanus, but this study contradicts the previous study [35] showed the place of delivery hadn't a significant effect on TT immunization. This might be the geographical location of the target population. Rural areas do have not sufficient infrastructure for hospitals and other health facilities, this leads to the place of delivery has an impact on immunization.

In this study, the occupation has no significant effect on women's protection from tetanus, but the other study contradicts this study [28] showed that occupation had a significant effect on women's protection from tetanus. This might be rural area mothers may have almost similar occupational activity, which leads to occupation having no significant effect on the immunization. Similarly, the age of the household head had no significant effect on women's protection from tetanus, which is also consistent with the other study [35]. The other variable sex of house old head hadn't significant effect with the odds of women's protected from tetanus. This might be in the rural area of the most African country house the old head is men and the most decision may be decided by men.

The odds of women being protected from tetanus were not affected by mothers who terminated the pregnancy. Finally, immunization costs covered by health insurance had not affected the odds of TT immunization, which inline the other study [35]. This might be most rural Africa countries' the cost of ANC visit was covered by the government. Therefore, large proportions of mother cost of immunization were covered by the government, and then it may be insignificant.

## Conclusion and recommendations

The percentage of mothers vaccinated with protective of TT in rural areas of East African countries was 50.4%. Vaccination coverage of rural women in Ethiopia (36.90%), Kenya (45.00%), Burundi (46.20%) and Malawi (48.7%) and Comoros (49.7%) was below 50% protected women against tetanus. However, Uganda (59.8%), Zimbabwe (60.2%), Zambia (51.40%), Tanzania (51.70), and Rwanda (65%) have a low prevalence of women with no tetanus protection. The study revealed that low vaccination with protective of TT immunization in the rural area east Africa countries was attributed by low educational level, long-distance from the health facility, poor economic status, older age of mother, number of ANC visit, and media exposure. The official visit of the ANC, the planned pregnancy, and the early beginning of the visit of the ANC should be emphasized. Likewise, it is recommended to raise awareness of rural women's education and to correct women's perceptions of the importance of immunization and the quality of TT service.

Furthermore, public health programs target rural mothers who are uneducated, poor households; longer distances from health facilities, mothers who have the problem of media

exposure, and mothers who have not used maternal health care services can be promoted for TT immunization. By enhancing vaccination against maternal tetanus, the government and other stakeholders should work adequately to increase vaccination against maternal tetanus.

## Limitation of the study

The study used secondary data from DHS in selected Africa countries. But the data have missing value in the variable and some variables are not found in some eastern Africa countries which are missing from the analysis. Therefore, missingness in the data was the main limitation of the study.

## Supporting information

**S1 File.**
(DOCX)

## Acknowledgments

We strongly recognize the DHS measuring program to provide access to DHS datasets in East Africa.

## Author Contributions

**Conceptualization:** Alebachew Taye Belay, Setegn Much Fenta, Mitiku Wale Muluneh.

**Data curation:** Alebachew Taye Belay, Setegn Much Fenta, Setegn Bayabil Agegn, Mitiku Wale Muluneh.

**Formal analysis:** Alebachew Taye Belay, Setegn Much Fenta, Setegn Bayabil Agegn.

**Funding acquisition:** Alebachew Taye Belay, Setegn Much Fenta, Setegn Bayabil Agegn.

**Investigation:** Alebachew Taye Belay, Setegn Much Fenta, Setegn Bayabil Agegn, Mitiku Wale Muluneh.

**Methodology:** Alebachew Taye Belay, Setegn Much Fenta, Setegn Bayabil Agegn.

**Project administration:** Alebachew Taye Belay, Setegn Much Fenta, Mitiku Wale Muluneh.

**Resources:** Alebachew Taye Belay, Setegn Bayabil Agegn.

**Software:** Alebachew Taye Belay, Setegn Much Fenta, Mitiku Wale Muluneh.

**Supervision:** Alebachew Taye Belay, Setegn Much Fenta, Setegn Bayabil Agegn.

**Validation:** Alebachew Taye Belay, Setegn Much Fenta, Setegn Bayabil Agegn, Mitiku Wale Muluneh.

**Visualization:** Alebachew Taye Belay, Setegn Bayabil Agegn, Mitiku Wale Muluneh.

**Writing – original draft:** Alebachew Taye Belay, Setegn Bayabil Agegn, Mitiku Wale Muluneh.

**Writing – review & editing:** Alebachew Taye Belay, Setegn Much Fenta, Mitiku Wale Muluneh.

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
