## [Decision Letter · Decision Letter 0]

15 Dec 2021

PONE-D-21-34591Prevalence and Risk Factors Associated with Rural Mother’s Protected against Tetanus in East Africa: Evidence from Demographic and health surveys of Nine East African countriesPLOS ONE

Dear Dr. Belay,

Thank you for submitting your manuscript to PLOS ONE. After careful consideration, we feel that it has merit but does not fully meet PLOS ONE’s publication criteria as it currently stands. Therefore, we invite you to submit a revised version of the manuscript that addresses the points raised during the review process.

The article has been carefully revised by two researchers from the fields who, while agreeing on the overall scientific merit, provide detailed feedback on issues to ammend regarding description of methods to ensure reproducibility and issues of interpretation of evidence.

We look forward to receiving your revised manuscript.

Kind regards,

José Antonio Ortega, Ph.D.

Academic Editor

PLOS ONE

Journal Requirements:

“No”

“No”

Reviewers' comments:

Reviewer's Responses to Questions

**Comments to the Author**

1. Is the manuscript technically sound, and do the data support the conclusions?

Reviewer #1: Yes

Reviewer #2: Yes

2. Has the statistical analysis been performed appropriately and rigorously? 

Reviewer #1: Yes

Reviewer #2: I Don't Know

3. Have the authors made all data underlying the findings in their manuscript fully available?

Reviewer #1: Yes

Reviewer #2: Yes

4. Is the manuscript presented in an intelligible fashion and written in standard English?

Reviewer #1: Yes

Reviewer #2: Yes

5. Review Comments to the Author

Reviewer #1: The research is of great relevance, bringing important results, especially with regard to the studied region, where the incidence of the disease is high, therefore, understanding the factors associated with vaccination in this region is relevant.

But there are some aspects to consider:

In the introduction, I suggest that you include a paragraph with information about the World Health Organization's strategies for preventing the disease in women in the at-risk area, we have an important document published in 2019 (Protecting all against tetanus-2019 by the World Health Organization https://www.who.int/publications/i/item/protecting-all-against-tetanus). In addition, I suggest that data on the incidence of the disease be updated, we now have more recent data.

Another aspect to be included is about the reach of the elimination of the disease in the studied region, was it achieved?

Also in the introduction, I suggest that the study hypothesis be included.

In the methods:

I suggest that a brief description of the outcome variable be made, as it was extracted from the Demographic and Health Survey (DHS), it was not clear to the reader. Also, describe what the tetanus vaccination questionnaire was like, was it based on any other questionnaire?

I suggest describing, in the paragraph "Variables of the study", how the immunization program specifically against tetanus works in the regions studied.

About the independent variables, how were they selected? Was it through a theoretical model?

The variable income, as the classification "Poor, middle and Rich" was considered, I suggest that you make a brief description.

Still, in methods, I suggest including information about ethical aspects of the research.

In results:

I suggest that in all tables, the sample number and year of data collection are included in the description.

In table 4, include the meaning of *

Also in table 4, I suggest that a column with the value of p be included.

Discussion:

I suggest that a systematic review and meta-analysis recently published on the subject be included:

Entitled "Tetanus vaccination in pregnant women: a systematic review and

meta-analysis of the global literature" https://pubmed.ncbi.nlm.nih.gov/34144334/

In addition, I would like to suggest to the authors the inclusion of a paragraph on the possible interference of tetanus vaccination during the pandemic, as we observe a drop in vaccination prevalence.

Still under discussion, I suggest that the limitations of the study be included.

Reviewer #2: Reviewer’s comments

Manuscript Title: Prevalence and Risk Factors Associated with Rural Mother’s Protected against Tetanus in East Africa: Evidence from Demographic and health surveys of Nine East African countries

Manuscript Number: PONE-D-21-34591

General comments

The authors assessed the prevalence and risk factors associated with protection against tetanus among mothers in rural settings in nine EA countries. The findings and recommendations from the article will add to the body of available literature on the subject. The article will be relevant to countries that are yet to eliminate maternal and neonatal tetanus as well as those are striving to sustain their elimination status

Minor comments

- In the background and other sections, the countries listed as “East African” do not all fall into that category, as Malawi, Zambia and Zimbabwe are considered as southern African countries, and Comoros as Island nation. The authors should consider using the appropriate categorizations for these countries

- In several sections of the manuscript and in tables, Comoros seems to have been replaced by Cameroon (A Central African country). The authors need to correct this error throughout the draft

- The authors need to read through the draft carefully to edit typos, and other errors that require editing

Major comments

Background

- It is important for the authors to indicate why Uganda, a key East African country is not included in the list despite having a 2016 DHS

- 34,019 (1%) deaths from NT in 2015 as against 3.3 m annual ND cannot be described as a “high number” The authors may wish to rephrase the sentence to “despite progress with global efforts to eliminate MNT, 34,019 neonatal deaths were attributed to tetanus in 2015”

- The statement “Maternal tetanus continues to be a major cause of neonatal and infant deaths in many developing countries” is not clear. I couldn't find such statements in the reference (Ref #6 & 7) cited. Please, ensure that references are correctly quoted. It is not clear how maternal tetanus causes neonatal and infant deaths. Maybe the authors meant “Neonatal tetanus”

- The statement “It is estimated that between 15,000 and 30,000 women die each year from tetanus acquired during or soon after pregnancy [5] would require an indication of the year of estimation

- Annual neonatal tetanus deaths were indicated for 2015 and 2017. The authors may wish to consider using the most recent estimates, which is 25,000 in 2018 (visit the link below). https://www.who.int/initiatives/maternal-and-neonatal-tetanus-elimination-(mnte)

- The statement “Similarly, the majority of the Sub-Saharan African countries could hardly reach the TT immunization target set to be covered [10]” requires that the authors indicate what the threshold TT coverage is referred to.

- I was unable to find the statement “Somalia, South Sudan, Afghanistan, Kenya, Nigeria, and Ethiopia reported the highest rates of neonatal tetanus mortality (1,000 deaths per 100,000 population) [11]” in the cited reference. Please, ensure to quote relevant references to support the statement

Methods

1. The countries selected are a mix of East African (Burundi, Kenya, Tanzania, Rwanda), Horn of Africa (Ethiopia), Island nation (Comoros) and southern African (Malawi, Zambia and Zimbabwe). The authors may wish to use the appropriate classification of countries. Other East African countries such as Uganda that had DHS conducted in 2016 would be nice to include. Kenya’s DHS was in 2014, and was wondering how it made the list of countries, if only DHS from 2016 were considered

2. While the DHS reports for the nine EA countries provide data on all variables for this paper, which should allow for comparison across countries or even at sub-national levels of the countries, no indications in this sections that comparisons will be made across countries or at subnational level. Moreover, Maternal and Neonatal tetanus elimination is based indicator performance at the district level.

3. If the paper relied on secondary data analyses, that should be stated clearly, as it will appear that there was extensive use of data already collected through the DHS.

4. While it appears that extensive scoping literature review was conducted especially around the factors that drive TT protection among women, this was not stated. The authors need to mention this.

Results

1. It is not clear what the cut-off TT threshold coverage for protection is. Please state this

2. “Cameroon (50.3% not protected and 49.7% protected)” – Cameroon is not in East Africa and is not part of the list in the other sections of the manuscript. Please, crosscheck. Probably the authors meant Comoros.

3. It is difficult to understand where percentage coverages quoted as “The prevalence of protected rural women from nine EA countries” come from. I looked up the TT2+ or Td2+ coverage figures reported through the most recent DHS reports in these countries and they do not seem to align with those reported in the manuscript. A clearer understanding in method section of how these figures were calculated would be useful.

4. From the method and result section, protection against tetanus is based on reported TT2+ or Td2+ coverage in the DHS. For several documented reasons, TT2+ method for assessing protection against tetanus among women of reproductive age (WRA) tends to grossly underestimate the true protection, hence the recommended use of the Protection at Birth (PAB), which you will notice in the DHS to be higher than TT2+ coverage in nearly all countries. For example, the 2016-2017 DHS for Burundi indicates 28.5% TT2+ coverage whereas PAB shows 84% coverage. It will be useful for the authors to consider including PAB coverage in this draft manuscript. Several reference materials are available on PAB methods.

5. In view of the wide disparities in health systems strength, demographics, development indices and economic status in the nine countries, it is difficult to understand why there were no efforts to compare findings across countries. While all the nine countries have been validated for the elimination of MNTE, several years back, the possibilities are there that disaggregating the data by country will show “high performing” and “low performing” countries. The authors may wish to disaggregate the data by country and consider indicating the ranges for the various variables, highlighting those with highest coverage and those with lowest coverage.

6. The statement “were significantly correlated with full childhood immunization (Table 2)” appears to be out of place. I am not sure why reference is being made to “full childhood immunization” here, since it neither the outcome nor one of the independent variables. Table 2 shows the correlation between TT protection amongst women and not full childhood immunization. Needs some explanation here or correct the sentence

Discussion

1. In the opening statement there seems to be an attempt to compare the findings from the nine “EA” countries lumped together with findings from single countries and even subnational levels of the countries. I am not sure that this is a good comparison. It will be more useful, if findings from EA are compared to those from West Africa, Central Africa or South Africa if such literatures exist.

2. Reference is required for the statement “This might be the target population of this study were women who live in the rural area of East Africa, which have a problem of early marriage in this area. Therefore, the age of the mother at first birth may relate to early marriage that causes a knowledge gap on ANC visits and TT protective immunization”

3. Ref #18 “Regarding the perception of distance to the health facility, nearly half (52%) of respondents perceived distance from the health facility as a big problem (Table 1)” is contrary to the citation by the authors of the same ref. Please, crosscheck and rectify.

Conclusions and recommendations

1. There appears to be many typos rendering the section unclear. The authors need to address these

2. The sweeping conclusion about TT protection and the influencing factors in EA may make it difficult for readers to understand the extent of the problem at individual country level.

Limitations

- There is no section that highlights the limitation of this study. The authors need to highlight the limitations

References

- The authors need to include URLs for all reference so readers can easily access and check the referenced materials

6. PLOS authors have the option to publish the peer review history of their article (what does this mean?). If published, this will include your full peer review and any attached files.

Reviewer #1: No

Reviewer #2: No

---

## [Author Response · Author response to Decision Letter 0]

25 Jan 2022

the comments raised by both reviewers were very interesting and important for the development of this article as well as for other work.

---

## [Decision Letter · Decision Letter 1]

10 Mar 2022

Prevalence and Risk Factors Associated with Rural Women’s Protected against Tetanus in East Africa: Evidence from Demographic and health surveys of Ten East African countries

PONE-D-21-34591R1

Dear Dr. Belay,

We’re pleased to inform you that your manuscript has been judged scientifically suitable for publication and will be formally accepted for publication once it meets all outstanding technical requirements.

Kind regards,

José Antonio Ortega, Ph.D.

Academic Editor

PLOS ONE

Additional Editor Comments (optional):

All suggestions have been addressed. Congratulations!

Reviewers' comments:

Reviewer's Responses to Questions

**Comments to the Author**

1. If the authors have adequately addressed your comments raised in a previous round of review and you feel that this manuscript is now acceptable for publication, you may indicate that here to bypass the “Comments to the Author” section, enter your conflict of interest statement in the “Confidential to Editor” section, and submit your "Accept" recommendation.

Reviewer #1: All comments have been addressed

Reviewer #2: All comments have been addressed

2. Is the manuscript technically sound, and do the data support the conclusions?

Reviewer #1: Yes

Reviewer #2: Yes

3. Has the statistical analysis been performed appropriately and rigorously? 

Reviewer #1: Yes

Reviewer #2: Yes

4. Have the authors made all data underlying the findings in their manuscript fully available?

Reviewer #1: Yes

Reviewer #2: Yes

5. Is the manuscript presented in an intelligible fashion and written in standard English?

Reviewer #1: Yes

Reviewer #2: Yes

6. Review Comments to the Author

Reviewer #1: The study is relevant because it is carried out in regions where vaccination rates are lower than expected, therefore, it presents important considerations for the literature. The authors included all requested suggestions.

Reviewer #2: While most of the concerns raised have been addressed by the authors, the revised draft appears to contain more typos and grammatical errors than earlier draft. This needs to be addressed for the final draft

7. PLOS authors have the option to publish the peer review history of their article (what does this mean?). If published, this will include your full peer review and any attached files.

Reviewer #1: No

Reviewer #2: No

---

## [Editor Report · Acceptance letter]

15 Mar 2022

PONE-D-21-34591R1 

Prevalence and Risk Factors Associated with Rural Women’s Protected against Tetanus in East Africa: Evidence from Demographic and health surveys of Ten East African countries 

Dear Dr. Belay:

I'm pleased to inform you that your manuscript has been deemed suitable for publication in PLOS ONE. Congratulations! Your manuscript is now with our production department. 

Kind regards, 

on behalf of

Dr. José Antonio Ortega 

Academic Editor

PLOS ONE